# "Long-term effects of center volume on transplant outcomes in adult kidney transplant recipients"

**Ambreen Azhar**[1], **Edem Defor**[2], **Dipankar Bandyopadhyay**[2], **Layla Kamal**[1], **Bekir Tanriover**[3], **Gaurav Gupta**[1] *

**1** Division of Nephrology, Department of Medicine, Virginia Commonwealth University, Richmond, VA, **2** Department of Biostatistics, Virginia Commonwealth University, Richmond, VA, **3** Division of Nephrology, Department of Medicine, College of Medicine, University of Arizona, Tucson, AZ

* Gaurav.gupta@vcuhealth.org

## Abstract

### Background

The influence of center volume on kidney transplant outcomes is a topic of ongoing debate. In this study, we employed competing risk analyses to accurately estimate the marginal probability of graft failure in the presence of competing events, such as mortality from other causes with long-term outcomes. The incorporation of immunosuppression protocols and extended follow-up offers additional insights. Our emphasis on long-term follow-up aligns with biological considerations where competing risks play a significant role.

### Methods

We examined data from 219,878 adult kidney-only transplantations across 256 U.S. transplant centers (January 2001-December 2015) sourced from the Organ Procurement and Transplantation Network registry. Centers were classified into quartiles by annual volume: low (Q1 = 28), medium (Q2 = 75), medium-high (Q3 = 121), and high (Q4 = 195). Our study investigated the relationship between center volume and 5-year outcomes, focusing on graft failure and mortality. Sub-population analyses included deceased donors, living donors, diabetic recipients, those with kidney donor profile index >85%, and re-transplants from deceased donors.

### Results

Adjusted cause-specific hazard ratios (aCHR) for Five-Year Graft Failure and Patient Death were examined by center volume, with low-volume centers as the reference standard (aCHR: 1.0). In deceased donors, medium-high and high-volume centers showed significantly lower cause-specific hazard ratios for graft failure (medium-high aCHR = 0.892, p<0.001; high aCHR = 0.953, p = 0.149) and patient death (medium-high aCHR = 0.828, p<0.001; high aCHR = 0.898, p = 0.003). Among living donors, no significant differences were found for graft failure, while a trend towards lower cause-specific hazard ratios for

**Data Availability Statement:** Data Availability: Our study used data from the United Network for Organ Sharing (UNOS) database, focusing on patients who underwent kidney transplantation between

2001 and 2015. The database can be accessed at
https://www.unos.org/data.

**Funding:** The author(s) received no specific
funding for this work.

**Competing interests:** The authors declare on
conflict of interest related to this work. G Gupta has
received honoraria from Alexion, CareDx,
Mallinckrodt; has served on the Scientific Advisory
Board for CareDx, Natera, Relypsa, Veloxis; and
has received research funding from Gilead and
Merck.

**Abbreviations:** KM, Kaplan–Meier; CRA,
Competing Risk Analysis; KDPI, Kidney Donor
Profile Index; OPTN, Organ Procurement and
Transplantation Network; BMI, Body Mass Index;
DM, Diabetes Mellitus; HTN, Hypertension; HCV,
Hepatitis C Virus Infection status; DCD, Donation
after Cardiac Death; CIT, Cold Ischemia Time; HCV,
Hepatitis C virus.

patient death was observed in medium-high (aCHR = 0.895, p = 0.107) and high-volume
centers (aCHR = 0.88, p = 0.061).

## Conclusion

Higher center volume is associated with significantly lower cause-specific hazard ratios for
graft failure and patient death in deceased donors, while a trend towards reduced cause-
specific hazard ratios for patient death is observed in living donors.

## Introduction

A major focus of research in the field of kidney transplantation is to address factors that can
compromise the overall goal of kidney transplantation. Transplant center volume is one such
factor that has been a topic of debate in the transplant community [1, 2]. Previous studies have
shown that high-volume kidney transplant centers had better short-term outcomes at 1 month
and 1 year post-transplant, presumably because of better peri-operative surgical technique and
management experience [1–3]. Transitions of care from transplant center to local nephrology
are variable from center to center and may be impacted by center volume with higher volume
centers transitioning patients earlier due to manpower limitations. The impact on outcomes of
specialized longitudinal medical care and the availability of a collaborative healthcare team,
which is more likely to be available at transplant center level rather than with community
nephrologists; has not been well studied [4]. While it has been shown that an increasing num-
ber of dialysis patients per nephrologist can have an adverse impact on outcomes, no such data
exists for kidney transplant patients being taken care of at transplant centers [5]. These prob-
lems are further complicated by a dearth of both the nephrology and the transplant nephrology
workforce in the United States [6, 7].

Sonnenberg et al categorized kidney transplant centers into low, medium, medium-high,
and high volume based on the mean annual number of kidney transplants from 2009 to 2013.
Their findings indicated no evidence of improved outcomes with increased center volume [8].
The study's limitations, including a focus on relatively short-term outcomes and reliance on
conventional survival analysis, may explain the lack of observed differences. The exclusion of
crucial factors such as immunosuppression protocols and long-term considerations could
have affected the detection of more nuanced effects on kidney transplant outcomes.

Recent publications suggest that conventional survival analysis methods ignoring the com-
peting event(s), such as Kaplan–Meier (KM) method and standard Cox proportional hazards
regression, may be inappropriate in the presence of competing risks, and alternative methods
specifically designed for analyzing competing risks data should be considered [9, 10]. This
problem deserves more attention in prognostic research in patients with advanced kidney dis-
ease, mainly because an important assumption of KM is that the censoring event (death or
graft loss) is an independent event with independent probability value [9]. It means that sub-
jects that are censored at a certain time point are expected to be as likely to experience the end-
point of interest as those that are not censored [9]. An alternative method used to estimate
survival probabilities is the competing risk analysis (CRA) which requires no assumption
about independence between the two risks, graft loss and death. In addition, Ters et al have
shown that factors like aging and diabetes that are generally associated with higher mortality
should be considered as competing events for death in kidney transplant patients while analyz-
ing long-term outcomes [11]. Their findings indicate that CRA could offer more precise

estimates of long-term graft survival and death, especially in subgroups of recipients where the impact of competing events is more significant.

This study aims to investigate the impact of kidney transplant center volume on graft loss and patient survival over 3- and 5-year post-transplant periods for adult kidney transplantations in the USA from 2001 to 2015. We implemented CRA guided by a natural biological rationale in this population, selecting a 5-year duration to uncover subtle outcomes over an extended timeframe with increased nuance. Our hypothesis posits that the transplant center's volume significantly influences clinical outcomes, affecting both graft failure and patient mortality. By pre-defining recipients at increased risk for separate subgroup analyses, including diabetic recipients, those receiving high kidney donor profile index (KDPI) deceased donor kidneys, and re-transplant recipients, we aim to unveil the nuanced biological dynamics of these outcomes, with CRA providing enhanced precision, particularly over an extended 5-year follow-up.

## Materials and methods

The United States National Organ Procurement Organization Network (OPTN) registry using Standard Transplant Analysis and Research (STAR) files was analyzed. Transplant center did not have access to personal information to identify the individual patients.

### Study population

We conducted a retrospective observational cohort study. All adult recipients (aged $\geq$ 18 years old) who received kidney only transplants from January 1, 2001, and December 31, 2015, were considered for analyses. Transplant centers whose total transplantation was less than 10 in the study period were not included.

**Exposure.** Transplant Center Volume: Our primary exposure of interest was the transplant center volume. Transplantation centers were categorized into quartiles (low, medium, medium-high, and high) based on their total volume (divided by years) of kidney-only transplants during the entire study period.

**Outcomes.** 5-Year Graft Failure and Mortality: Our primary outcomes were graft failure and mortality within 5 years post-transplant. Graft failure was defined as a return to dialysis or re-transplantation. Data was censored on December 31, 2020, to allow a five-year follow-up for the study cohort.

**Competing events analysis.** Patient Death vs. Graft Failure: Our competing outcomes of interest involved the interplay between patient death and graft failure in the context of kidney transplantation. Graft failure, defined as a return to dialysis or re-transplantation, and patient death were considered as competing measures. Specifically, a patient death was deemed a competing measure if the STAR file contained a record of a date of death alongside a record of a functioning graft. Additionally, it was considered a competing measure when there was no graft failure date or a date indicating resumption of maintenance dialysis reported. This analysis aims to understand the dynamic relationship between these two critical outcomes in the study cohort.

### Statistical analyses

Outcomes across quartiles were described as frequencies with percentages or means with standard deviations where appropriate and compared using Chi-squared tests and Kruskal-Wallis tests, respectively. To analyze the effects of center volume on kidney graft failure, we used a competing risks regression approach since our competing event could preclude individuals from reaching the clinical endpoint. Due to the nature of our study being one of etiology rather

than in predicting an individual's risk for an outcome [12], we particularly carried out a cause-specific proportional hazards frailty model, clustering on transplant center, for each outcome.

It is noteworthy that the risk set in a cause-specific hazards CRA includes only subjects who have not yet experienced any event. The estimated regression parameters thus directly quantify the cause-specific hazard ratios among those individuals who are at risk of developing the event of interest [13]. This is distinct from the Fine and Gray sub-distribution hazards CRA model, which includes subjects who are currently event free and those who have experienced the competing event [9]. We fitted our cause-specific hazards model using the "coxph" function in the "survival" package in R. In the model fit; we treated the competing event as censored for each event type. Hence separate Cox regression models were used to study the event of interest.

We performed a pre-specified subgroup analysis for patients that may have a presumed high risk of allograft failure based on donor and recipient characteristics. Analyses were conducted under various subset settings: (1) LDKT, (2) DDKT, (3) transplants for diabetic recipients, (4) high KDPI transplants (≥85% KDPI), and (5) those with re-transplants. Immunosuppression protocols were recorded as T-cell depletion (anti-thymocyte globulin or Alemtuzumab), IL2-Receptor Depletion (Basiliximab or Daclizumab) for induction. Maintenance immunosuppression was recorded as drugs used at the time of patient discharge.

All multivariable models were adjusted for the following covariates: recipient age, sex, race, body mass index, diabetes status, panel reactive antibody (PRA) levels, presence of hepatitis C virus antibody positivity at transplant (HCV), wait-time pre-transplant, dialysis (yes/no) at transplant status, length of stay for index transplant hospitalization, treated for kidney rejection within 1 year, donor age, gender, race, and post-transplant immunosuppression. Donor factors including sex, age, race, diabetes, hypertension, HCV antibody status, donor terminal creatinine and donation after cardiac death (DCD) status. Living donor transplant outcomes were adjusted for recipient's age, sex, race, body mass index (BMI), diabetes mellitus (DM), cause of renal disease, prior solid organ transplantation, pre-transplant dialysis, waitlist days and panel reactive antibody levels (PRA). HLA mismatch was a covariate for both living and DDKT. We identified the covariates for adjustment based on the clinical judgement and previously available literature [14–23].

## Results

The final cohort included 219,878 kidneys only transplantations at 256 transplant centers (**Table 1**). Mean annual number of kidney transplants were 2–65, 66–110, 111–195 and 198–315 per year at low (Q1), medium (Q2), medium-high (Q3) and high volume (Q4) transplant centers respectively.

### DDKT recipients

There was a slight variation in the recipient characteristics among the four groups (**Table 2**). Cold ischemia time was relatively higher in Q3 and Q4 centers compared with the other groups (~18–19 hours vs ~16–17 hours; $p<0.001$). Interestingly, the length of hospital stay was lowest in the medium-high (Q3 group; 6.6 days) compared with the other groups (~7–8 days; $p<0.001$) (**Table 2**). Q3 (61.2%) and Q4 (59.37%) centers ($p<0.001$) were more likely to use induction T-cell depletion compared with Q1 (52.34%) and Q2 (54.44%) centers. Steroid based maintenance regimens were used more in low volume centers as compared to high volume centers (Q1 = 90.15%, Q2 = 88.53%, Q3 = 91.37%, Q4 = 84.55%; $p< 0.001$).

**Table 1. Characteristics of adult kidney transplant recipients for study period of 2001–2015.**

| | Overall (n, %) | Low | Medium | Medium-High | High | p-value [*] |
|---|---|---|---|---|---|---|
| Total no. of kidney transplantations | 219,878 | 54,491 (24.8%) | 56,464 (25.7%) | 56,362 (25.6%) | 52,561 (23.9%) | <0.001 |
| Deceased donor recipients | 135,505 (61.6%) | 35,757 (65.6%) | 36,696 (65.0%) | 35,673 (63.3%) | 27,379 (52.1%) | <0.001 |
| Re-transplant (>1) recipients | 17,760 (13.1%) | 4,260 (11.9%) | 4,786 (13.0%) | 4,678 (13.1%) | 4,036 (14.7%) | <0.001 |
| High-KDPI (>85) recipients | 10,225 (7.5%) | 2,098 (5.9%) | 2,851 (7.8%) | 3,013 (8.4%) | 2,263 (8.3%) | <0.001 |
| Living donor recipients | 84, 373 (38.4%) | 18,734 (34.4%) | 19,768 (35.0%) | 20, 689 (36.7%) | 25, 182 (47.9%) | <0.001 |
| Diabetic recipients | 68,865 (31.3%) | 17,888 (32.8%) | 17,437 (30.9%) | 17,429 (30.9%) | 16,111 (30.6%) | <0.001 |
| No. of centers | 256 | 157 | 51 | 30 | 18 | - |
| Total center volume | 10–4398 | 10–876 | 884–1412 | 1415–2534 | 2536–4398 | - |
| Median (Range) annual volume [a] | 47 (1–293) | 28 (1–58) | 75 (59–94) | 121 (94–168) | 195 (169–293) | - |

Notes:

[*] Testing the difference between all four groups;

[a] Range is total volume divided by study period (15 years)

Abbreviations: KDPI, Kidney Donor Profile Index

Donor characteristics (Table 2) varied across the volume-based quartiles with a graded increase in the utilization of high KDPI kidneys (Q1: 1.21, Q2: 1.25; Q3: 1.27 and Q4: 1.28; $p<0.001$).

## LDKT recipients

More living donor transplants were done at high volume centers (Q4 = 47.9%, vs Q3 = 36.7%, Q2 = 35.0%, Q1 = 34.4%). Similarly, more pre-emptive living donor transplants were done at Q4 centers where ~65% patients were on dialysis at the time of transplant as compared to Q1 centers where ~70% patients were on dialysis. Q4 centers did more re-transplants (12.68%) than Q1 centers (8.75%). There were substantial immunosuppressive regimen differences with significantly ($p<0.001$) more T-cell depletion use in Q4 (55.8%) vs Q3 (50.39%), Q2 (41.3%) and Q1 (42.41%) centers and significant less CNI (Q4 = 56.37%, Q3 = 71.19%, Q2 = 75.12%, Q1 = 71.57%) and maintenance steroid use (Q4 = 83.79%, Q3 = 87.52%, Q2 = 88.86%, Q1 = 89.79%) in Q4 centers vs Q3, Q2 and Q1 centers. The length of hospital stay was lowest in the medium-high (Q3 group; 5.4 days) compared with the other groups (5.8–6.4 days; p<0.001).

## Diabetic recipients

There were 68,865 (31%) diabetic kidney transplant recipients in our cohort with a relatively even proportion of transplants across the four quartiles (Table 1). Table 4 shows overall diabetic cohort characteristics and S3 and S4 Tables in S1 Appendix shows outcomes specifically focusing on graft failure and patient death in diabetic patients of living and deceased donors.

In the analysis of living donors among diabetic patients, the cause-specific hazard ratios (aCHR) for graft failure suggest that there is no statistically significant association with center volume. Moving on to deceased donors within the diabetic patient group, the aCHR for graft failure indicate a significant association for medium-high volume (aCHR = 0.867, p = 0.024) compared to low volume. However, for patient death, there is no significant association for any volume group. Factors including recipient age, ethnicity, BMI, peak PRA, and LOS are significantly linked to graft failure and patient death in both living and deceased donors. Additionally, donor variables such as age, ethnicity, BMI, height, and diabetes influence outcomes.

**Table 2. Characteristics for deceased donor adult kidney transplant recipients.**

| | Overall (n = 135505) | Low (Q1; n = 35757) | Medium (Q2; n = 36696) | Medium-High (Q3; n = 35673) | High (Q4; n = 27379) | P-value [*] |
|---|---|---|---|---|---|---|
| **Recipient Characteristics** | | | | | | |
| Recipient Age, Years | 51.91 ± 13.18 | 51.77 ± 13.41 | 51.94 ± 13.21 | 51.71 ± 12.99 | 52.34 ± 13.10 | <0.001 |
| Recipient Sex, Male | 81559 (60.19%) | 22063 (61.70%) | 21937 (59.78%) | 21136 (59.25%) | 16423 (59.98%) | <0.001 |
| AA Recipient Ethnicity | 43351 (32.18%) | 9891 (27.87%) | 12638 (34.61%) | 12511 (35.23%) | 8311 (30.54%) | <0.001 |
| Missing | 771 (0.6%) | 265 (0.7%) | 182 (0.5%) | 159 (0.4%) | 165 (0.6%) | |
| Recipient BMI | 27.83 ± 5.43 | 27.90 ± 5.42 | 27.98 ± 5.49 | 27.63 ± 5.28 | 27.80 ± 5.56 | <0.001 |
| Missing | 830 (0.6%) | 123 (0.3%) | 149 (0.4%) | 150 (0.4%) | 408 (1.5%) | |
| Recipient Diabetes, Y | 45162 (33.39%) | 12389 (34.71%) | 12021 (32.82%) | 11878 (33.36%) | 8874 (32.49%) | <0.001 |
| Missing | 256 (0.2%) | 62 (0.2%) | 65 (0.2%) | 65 (0.2%) | 64 (0.2%) | |
| Recipient Peak PRA | 19.59 ± 33.10 | 17.91 ± 31.89 | 19.52 ± 33.06 | 22.25 ± 34.77 | 18.42 ± 32.28 | <0.001 |
| Missing | 2545 (1.9%) | 379 (1.1%) | 558 (1.5%) | 862 (2.4%) | 746 (2.7%) | |
| Recipient Hepatitis C Antibody, Positive | 7677 (6.06%) | 1987 (5.80%) | 1965 (5.77%) | 2082 (6.15%) | 1643 (6.71%) | <0.001 |
| Missing | 8895 (6.6%) | 1508 (4.2%) | 2647 (7.2%) | 1840 (5.2%) | 2900 (10.6%) | |
| Recipient Dialysis at TX, Y | 121640 (90.32%) | 32529 (91.30%) | 33076 (90.89%) | 31748 (89.34%) | 24287 (89.53%) | <0.001 |
| Missing | 823 (0.6%) | 129 (0.4%) | 304 (0.8%) | 137 (0.4%) | 253 (0.9%) | |
| Recipient Kidney CIT | 17.84 ± 8.47 | 16.79 ± 7.53 | 16.99 ± 7.51 | 18.38 ± 8.74 | 19.60 ± 9.94 | <0.001 |
| Missing | | | | | | |
| HLA antigen mismatch | 3.90 ± 1.74 | 3.89 ± 1.74 | 3.96 ± 1.70 | 3.86 ± 1.74 | 3.88 ± 1.76 | <0.001 |
| Missing | 28 (<0.1%) | 5 (<0.1%) | 15 (<0.1%) | 6 (<0.1%) | 2 (<0.1%) | |
| Re-transplant, Y | 17760 (13.11%) | 4260 (11.91%) | 4786 (13.04%) | 4678 (13.11%) | 4036 (14.74%) | <0.001 |
| Immunosuppression | | | | | | |
| T-cell Depletion Induction | 76781 (56.66%) | 18715 (52.34%) | 19979 (54.44%) | 21833 (61.20%) | 16254 (59.37%) | <0.001 |
| IL-2RA Induction | 29891 (22.06%) | 9964 (27.87%) | 6818 (18.58%) | 7250 (20.32%) | 5859 (21.40%) | <0.001 |
| Calcineurin Inhibitor | 120465 (88.90%) | 32235 (90.15%) | 32486 (88.53%) | 32596 (91.37%) | 23148 (84.55%) | <0.001 |
| Maintenance Steroids | 99330 (73.30%) | 27271 (76.27%) | 28101 (76.58%) | 27272 (76.45%) | 16686 (60.94%) | <0.001 |
| **Donor Characteristics** | | | | | | |
| Donor Age, Years | 38.29 ± 16.60 | 37.69 ± 16.21 | 38.46 ± 16.51 | 38.00 ± 16.93 | 39.20 ± 16.77 | <0.001 |
| Donor Sex, Male | 81069 (59.83%) | 21407 (59.87%) | 22115 (60.27%) | 21332 (59.80%) | 16215 (59.22%) | 0.068 |
| AA Donor Ethnicity | 17694 (13.16%) | 3767 (10.58%) | 5143 (14.21%) | 5311 (14.97%) | 3473 (12.76%) | <0.001 |
| Missing | 1035 (0.8%) | 158 (0.4%) | 513 (1.4%) | 199 (0.6%) | 165 (0.6%) | |
| Donor BMI | 27.01 ± 6.70 | 26.83 ± 6.45 | 27.06 ± 6.60 | 27.04 ± 6.87 | 27.13 ± 6.93 | <0.001 |
| Missing | 224 (0.2%) | 61 (0.2%) | 64 (0.2%) | 47 (0.1%) | 52 (0.2%) | |
| Donor Diabetes, Y | 9106 (6.75%) | 2055 (5.77%) | 2421 (6.63%) | 2610 (7.35%) | 2020 (7.42%) | <0.001 |
| Missing | 625 (0.5%) | 123 (0.3%) | 167 (0.5%) | 182 (0.5%) | 153 (0.6%) | |
| Donor Hypertension, Y | 35636 (26.47%) | 8579 (24.13%) | 9886 (27.11%) | 9563 (27.01%) | 7608 (27.97%) | <0.001 |
| Missing | 887 (0.7%) | 209 (0.6%) | 228 (0.6%) | 273 (0.8%) | 177 (0.6%) | |
| Donor Hep C Antibody, Positive | 3442 (2.54%) | 723 (2.03%) | 941 (2.57%) | 847 (2.38%) | 931 (3.41%) | <0.001 |
| Missing | 229 (0.2%) | 72 (0.2%) | 56 (0.2%) | 59 (0.2%) | 42 (0.2%) | |
| Donor Don. After Cardiac Death, Y | 16609 (12.26%) | 4413 (12.34%) | 4886 (13.31%) | 4117 (11.54%) | 3193 (11.66%) | <0.001 |
| Missing | 2 (<0.1%) | 0 (0%) | 0 (0%) | 0 (0%) | 2 (<0.1%) | |
| Donor Terminal Creatinine | 1.13 ± 0.93 | 1.05 ± 0.81 | 1.10 ± 0.89 | 1.21 ± 1.06 | 1.15 ± 0.96 | <0.001 |
| Missing | 79 (0.1%) | 19 (0.1%) | 21 (0.1%) | 19 (0.1%) | 20 (0.1%) | |
| **Outcomes** | | | | | | |
| Graft failure (5 yr) | 19465 (14.36%) | 4987 (13.95%) | 5494 (14.97%) | 4928 (13.81%) | 4056 (14.81%) | <0.001 |
| Death (5 yr) | 14670 (10.83%) | 4117 (11.51%) | 4005 (10.91%) | 3466 (9.72%) | 3082 (11.26%) | <0.001 |
| Recipient LOS, Days | 7.72 ± 11.96 | 8.36 ± 11.33 | 7.74 ± 9.96 | 6.66 ± 9.81 | 8.25 ± 16.73 | <0.001 |

*(Continued)*

**Table 2.** (Continued)

| | Overall (n = 135505) | Low (Q1; n = 35757) | Medium (Q2; n = 36696) | Medium-High (Q3; n = 35673) | High (Q4; n = 27379) | P-value [*] |
|---|---|---|---|---|---|---|
| Missing | 389 (0.3%) | 107 (0.3%) | 74 (0.2%) | 97 (0.3%) | 111 (0.4%) | |
| Follow-up (mean±SD; days) | 2322.37 ± 1518.63 | 2289.26 ± 1543.24 | 2352.24 ± 1514.60 | 2339.10 ± 1475.11 | 2303.76 ± 1546.25 | <0.001 |
| Treated for rejection within 1 year | 10820 (10.54%) | 2990 (10.77%) | 2904 (10.59%) | 2711 (9.83%) | 2215 (11.14%) | <0.001 |
| Missing | 32856 (24.2%) | 7991 (22.3%) | 9286 (25.3%) | 8086 (22.7%) | 7493 (27.4%) | |

Notes: Continuous variables summarized as mean±standard deviation; categorical variables, as count (percentage).

"*" Testing the difference between all four groups.

Abbreviations: AA, African American; BMI, Body mass index; CIT, Cold ischemia time; LOS, Length of stay; TX, Transplant; IL-2RA, Interleukin-2 receptor antagonist; HLA, Human leukocyte antigen; PRA, Panel reactive antibody; Y, Yes

These observations imply that the impact of center volume on outcomes is more notable in deceased donor kidney transplants for diabetic patients than in living donor transplants.

## High KDPI recipients

As expected, the recipients of these kidneys were older at an average of 60 years of age (S1 Table in S1 Appendix). Of the 10, 225 high KDPI (>85%) transplants performed, a higher proportion (~30% each) were performed at Q2 and Q3 centers vs the other two quartiles. Length of hospital stay was also the lowest in the medium-high (Q3 group; 6.9) compared with the other groups (8.6–9.7 days; p<0.001). Q4 centers were again less likely (p<0.001) to use maintenance CNI (80.73% vs ~86–91%) and/or maintenance steroids (54.88% vs ~70–74%) compared with the other three quartiles.

## Re-transplant recipients

S2 Table in S1 Appendix presents the characteristics of these patients. A total of 17,760 patients were included. The peak PRA for these patients was high (~55–60%) with the highest PRA in the Q3 group. Similar trends as before with more T-cell depletion use in Q4 (72.72%) and Q3 (74.75%), vs Q2 (63.96%) and Q1 (65.02%) centers; and significant less CNI (Q4 = 84.76%, Q3 = 90.77%, Q2 = 87.67%, Q1 = 89.32%) and maintenance steroid use (Q4 = 69.67%, Q3 = 84.84%, Q2 = 82.49%, Q1 = 82.18%) in Q4 centers vs Q3, Q2 and Q1 centers. Length of hospital stay was once again lowest in the medium-high (Q3 group; 7 days) compared with the other groups (8–8.8days; p<0.001).

## Transplant and patient outcomes

Fig 1 show unadjusted survival curves for patient mortality attributable (or not) to graft loss. Absolute 5-year survival rates based upon sub-group analyses are also depicted in Tables 2–4, S1 and S2 Tables in S1 Appendix. Overall, both 5-year graft loss and death rates were lowest for Q3 (medium-high) centers compared with the other three quartiles for all sub-populations.

Fig 2 show the hazards of graft loss and patient death stratified by center volume using sub-population based CRA models. S3 Table in S1 Appendix shows the competing risk regression model for graft loss and patient death after adjustment for covariates when considering all DDKT. Q1 centers were considered the reference. Medium-high (Q3) centers had the lowest adjusted cause-specific hazard ratio for graft loss [aCHR: 0.892 (95% CI: 0.841–0.946)] and patient death [aCHR: 0.828 (95%CI: 0.776–0.884)]. A similar but weaker trend was seen for

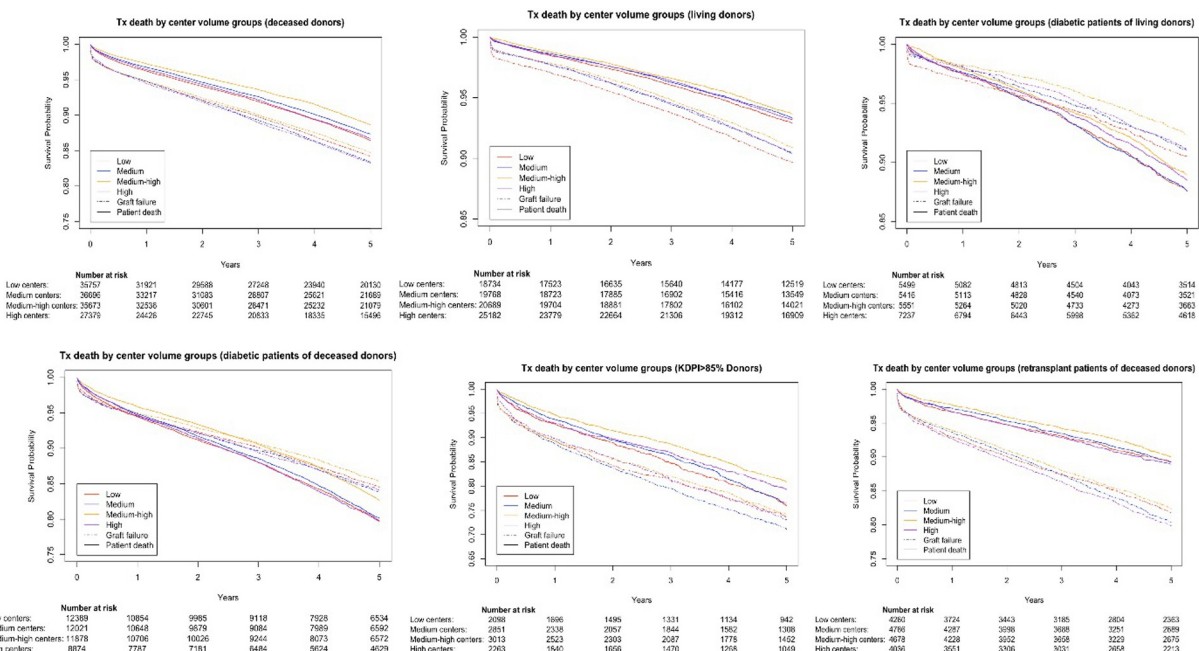

**Fig 1.** Cause-specific survival rates of patient death (solid lines) not from kidney graft failure and graft failure (dotted lines) by center volume groups (low, medium, medium-high and high-volume centers) for: **(A) Deceased donors**: Five-Year Graft Survival was 84.15% for low-volume, 83.36% for medium volume, 84.73% for medium-high volume, and 83.25% for high-volume centers. Five-Year Patient Survival was 86.43% for low-volume, 87.33% for medium volume, 88.65% for medium-high volume, and 86.71% for high-volume centers. **(B) Living donors**: Five-Year Graft Survival was 89.68% for low-volume, 90.39% for medium volume, 90.89% for medium-high volume, and 90.48% for high-volume centers. Five-Year Patient Survival was 92.89% for low-volume, 93.30% for medium volume, 93.66% for medium-high volume, and 93.17% for high-volume centers. **(C) Diabetic patients**: Five-Year Graft Survival was 86.44% for low-volume, 86.11% for medium volume, 87.59% for medium-high volume, and 87.32% for high-volume centers. Five-Year Patient Survival was 82.25% for low-volume, 82.55% for medium volume, 84.75% for medium-high volume, and 83.75% for high-volume centers. **(D) KDPI > 85%:** Five-Year Graft Survival was 73.52% for low-volume, 71.08% for medium volume, 74.13% for medium-high volume, and 73.04% for high-volume centers. Five-Year Patient Survival was 75.97% for low-volume, 76.60% for medium volume, 80.91% for medium-high volume, and 79.31% for high-volume centers. **(E) Re-transplants**: Five-Year Graft Survival was 81.76% for low-volume, 80.43% for medium volume, 82.33% for medium-high volume, and 79.88% for high-volume centers. Five-Year Patient Survival was 89.21% for low-volume, 89.25% for medium volume, 89.98% for medium-high volume, and 88.95% for high-volume centers.

Q4 centers. Several other biologically plausible factors, like older recipient age, diabetes, male sex, African American race etc. were confirmed to have adverse impact on outcomes.

S4 Table in S1 Appendix shows the competing risk regression model for graft loss and patient death after adjustment for covariates when considering all living donor kidney transplants. Q1 centers were considered the reference. Although both Q3 and Q4 centers had a numerically lower adjusted hazards for graft loss and death, these differences did not reach statistical significance.

CRA analyses for diabetic recipients are presented in S5 Table in S1 Appendix. Once again, Q3 centers had the lowest hazards of graft loss (aCHR: 0.848) and patient death (aCHR: 0.826). S6 and S7 Tables in S1 Appendix show the sub-population CRA analyses for KDPI>85% transplants and re-transplants. Adjusted hazards for graft outcomes were largely similar across the four quartiles in these subgroup analyses. The hazards of patient death were lowest for Q3 centers (Fig 2).

## Discussion

In this large retrospective observational study using a novel statistical competing risks design, we find that transplant center volume has an independent effect on both long-term graft loss

**Table 3. Characteristics for living donor adult kidney transplant recipients.**

| | Overall (n = 84373) | Low (Q1; n = 18734) | Medium (Q2; n = 19768) | Medium-High (Q3; n = 20689) | High (Q4; n = 25182) | P-value [*] |
|---|---|---|---|---|---|---|
| **Recipient Characteristics** | | | | | | |
| Recipient Age, Years | 47.35 ± 13.95 | 47.03 ± 14.25 | 47.14 ± 13.82 | 47.16 ± 13.89 | 47.90 ± 13.86 | <0.001 |
| Recipient Sex, Male | 51225 (60.71%) | 11645 (62.16%) | 12025 (60.83%) | 12575 (60.78%) | 14980 (59.49%) | <0.001 |
| AA Recipient Ethnicity | 11769 (14.03%) | 2364 (12.72%) | 3311 (16.86%) | 2904 (14.09%) | 3190 (12.73%) | <0.001 |
| Missing | 482 (0.6%) | 156 (0.8%) | 126 (0.6%) | 77 (0.4%) | 123 (0.5%) | |
| Recipient BMI | 27.30 ± 5.46 | 27.25 ± 5.34 | 27.44 ± 5.54 | 26.97 ± 5.23 | 27.52 ± 5.65 | <0.001 |
| Missing | 1126 (1.3%) | 121 (0.6%) | 243 (1.2%) | 142 (0.7%) | 620 (2.5%) | |
| Recipient Diabetes, Y | 23703 (28.10%) | 5499 (29.36%) | 5416 (27.40%) | 5551 (26.84%) | 7237 (28.74%) | <0.001 |
| Missing | 20 (<0.1%) | 6 (<0.1%) | 5 (<0.1%) | 4 (<0.1%) | 5 (<0.1%) | |
| Recipient Peak PRA | 9.91 ± 23.08 | 8.77 ± 21.63 | 8.95 ± 21.76 | 11.45 ± 24.40 | 10.20 ± 23.84 | <0.001 |
| Missing | 19804 (23.5%) | 4764 (25.4%) | 4766 (24.1%) | 4344 (21.0%) | 5930 (23.5%) | |
| Recipient Hepatitis C Antibody, Positive | 2237 (2.82%) | 496 (2.76%) | 523 (2.81%) | 671 (3.48%) | 547 (2.34%) | <0.001 |
| Missing | 5186 (6.1%) | 774 (4.1%) | 1126 (5.7%) | 1434 (6.9%) | 1852 (7.4%) | |
| Recipient Dialysis at TX, Y | 56454 (67.65%) | 13044 (69.90%) | 13513 (69.33%) | 13802 (67.02%) | 16095 (65.15%) | <0.001 |
| Missing | 923 (1.1%) | 73 (0.4%) | 276 (1.4%) | 95 (0.5%) | 479 (1.9%) | |
| HLA antigen mismatch | 3.31 ± 1.68 | 3.27 ± 1.67 | 3.28 ± 1.69 | 3.32 ± 1.68 | 3.35 ± 1.68 | <0.001 |
| Missing | 687 (0.8%) | 323 (1.7%) | 177 (0.9%) | 84 (0.4%) | 103 (0.4%) | |
| Re-transplant, Y | 8829 (10.46%) | 1639 (8.75%) | 1963 (9.93%) | 2035 (9.84%) | 3192 (12.68%) | <0.001 |
| Immunosuppression | | | | | | |
| T-cell Depletion Induction | 40603 (48.12%) | 7945 (42.41%) | 8167 (41.31%) | 10425 (50.39%) | 14066 (55.86%) | <0.001 |
| IL-2RA Induction | 24174 (28.65%) | 6448 (34.42%) | 6292 (31.83%) | 5393 (26.07%) | 6041 (23.99%) | <0.001 |
| Calcineurin Inhibitor | 73594 (87.22%) | 16822 (89.79%) | 17565 (88.86%) | 18107 (87.52%) | 21100 (83.79%) | <0.001 |
| Maintenance Steroids | 57180 (67.77%) | 13408 (71.57%) | 14850 (75.12%) | 14728 (71.19%) | 14194 (56.37%) | <0.001 |
| **Donor Characteristics** | | | | | | |
| Donor Age, Years | 41.33 ± 11.49 | 41.11 ± 11.28 | 41.02 ± 11.55 | 41.01 ± 11.43 | 41.99 ± 11.63 | <0.001 |
| Donor Sex, Male | 33315 (39.49%) | 7363 (39.30%) | 7933 (40.13%) | 8091 (39.11%) | 9928 (39.42%) | 0.173 |
| AA Donor Ethnicity | 10304 (12.29%) | 2081 (11.19%) | 2918 (14.88%) | 2554 (12.41%) | 2751 (10.99%) | <0.001 |
| Missing | 545 (0.6%) | 136 (0.7%) | 158 (0.8%) | 109 (0.5%) | 142 (0.6%) | |
| Donor BMI | 26.90 ± 4.40 | 26.89 ± 4.34 | 26.98 ± 4.43 | 26.66 ± 4.23 | 27.04 ± 4.54 | <0.001 |
| Missing | 5117 (6.1%) | 727 (3.9%) | 1462 (7.4%) | 925 (4.5%) | 2003 (8.0%) | |
| Donor Hypertension, Y | 1708 (2.05%) | 362 (1.94%) | 425 (2.16%) | 373 (1.84%) | 548 (2.20%) | 0.021 |
| Missing | 876 (1.0%) | 55 (0.3%) | 100 (0.5%) | 410 (2.0%) | 311 (1.2%) | |
| **Outcomes** | | | | | | |
| Graft failure (5 yr) | 7351 (8.71%) | 1753 (9.36%) | 1728 (8.74%) | 1711 (8.27%) | 2159 (8.57%) | 0.001 |
| Death (5 yr) | 5012 (5.94%) | 1165 (6.22%) | 1177 (5.95%) | 1160 (5.61%) | 1510 (6.00%) | 0.076 |
| Recipient LOS, Days | 6.03 ± 13.02 | 6.43 ± 8.92 | 5.88 ± 7.58 | 5.41 ± 12.15 | 6.37 ± 18.49 | <0.001 |
| Missing | 136 (0.2%) | 28 (0.1%) | 28 (0.1%) | 25 (0.1%) | 55 (0.2%) | |
| Follow-up (days) | 2757.45 ± 1610.11 | 2730.62 ± 1636.99 | 2820.60 ± 1628.12 | 2784.92 ± 1605.69 | 2705.27 ± 1576.94 | <0.001 |
| Treated for rejection within 1 year | 6752 (10.23%) | 1572 (10.53%) | 1667 (10.86%) | 1699 (10.29%) | 1814 (9.45%) | <0.001 |
| Missing | 18394 (21.8%) | 3802 (20.3%) | 4417 (22.3%) | 4183 (20.2%) | 5992 (23.8%) | |

Notes: Continuous variables summarized as mean±standard deviation; categorical variables, as count (percentage).

"*" Testing the difference between all four groups.

Abbreviations: AA, African American; BMI, Body mass index; CIT, Cold ischemia time; LOS, Length of stay; TX, Transplant; IL-2RA, Interleukin-2 receptor antagonist; HLA, Human leukocyte antigen; PRA, Panel reactive antibody; Y, Yes

**Table 4. Characteristics of diabetic adult kidney transplant recipients of living or deceased donor kidneys.**

| | Overall (n = 68865) | Low (Q1; n = 17888) | Medium (Q2; n = 17437) | Medium-High (Q3; n = 17429) | High (Q4; n = 16111) | P-value [*] |
|---|---|---|---|---|---|---|
| **Recipient Characteristics** | | | | | | |
| Recipient Age, Years | 55.78 ± 10.74 | 55.77 ± 10.73 | 56.08 ± 10.70 | 55.78 ± 10.57 | 55.46 ± 10.97 | <0.001 |
| Recipient Sex, Male | 44436 (64.53%) | 11805 (65.99%) | 11119 (63.77%) | 11215 (64.35%) | 10297 (63.91%) | <0.001 |
| AA Recipient Ethnicity | 17963 (26.24%) | 3948 (22.27%) | 5267 (30.37%) | 5057 (29.12%) | 3691 (23.04%) | <0.001 |
| Recipient BMI | 29.21 ± 5.31 | 29.18 ± 5.22 | 29.44 ± 5.38 | 28.88 ± 5.09 | 29.37 ± 5.52 | <0.001 |
| Missing | 471 (0.7%) | 47 (0.3%) | 87 (0.5%) | 103 (0.6%) | 234 (1.5%) | |
| Recipient Peak PRA | 13.79 ± 28.08 | 12.86 ± 27.17 | 13.96 ± 28.41 | 15.85 ± 29.71 | 12.35 ± 26.65 | <0.001 |
| Missing | 6468 (9.4%) | 1608 (9.0%) | 1534 (8.8%) | 1427 (8.2%) | 1899 (11.8%) | |
| Recipient Kidney CIT | 13.37 ± 10.52 | 12.93 ± 9.52 | 13.06 ± 9.55 | 14.52 ± 10.96 | 12.95 ± 11.96 | <0.001 |
| Missing | 8015 (11.6%) | 1771 (9.9%) | 1996 (11.4%) | 1999 (11.5%) | 2249 (14.0%) | |
| Recipient Hepatitis C Antibody, Positive | 3643 (5.63%) | 893 (5.18%) | 927 (5.65%) | 1013 (6.16%) | 810 (5.53%) | 0.001 |
| Missing | 4144 (6.0%) | 637 (3.6%) | 1042 (6.0%) | 991 (5.7%) | 1474 (9.1%) | |
| Recipient Dialysis at TX, Y | 58370 (85.35%) | 15524 (87.04%) | 15069 (87.18%) | 14698 (84.70%) | 13079 (82.19%) | <0.001 |
| Missing | 479 (0.7%) | 53 (0.3%) | 152 (0.9%) | 76 (0.4%) | 198 (1.2%) | |
| HLA antigen mismatch | 3.73 ± 1.72 | 3.73 ± 1.72 | 3.79 ± 1.70 | 3.73 ± 1.72 | 3.66 ± 1.74 | <0.001 |
| Missing | 198 (0.3%) | 110 (0.6%) | 44 (0.3%) | 18 (0.1%) | 26 (0.2%) | |
| Re-transplant, Y | 5275 (7.66%) | 1158 (6.47%) | 1275 (7.31%) | 1355 (7.77%) | 1487 (9.23%) | <0.001 |
| **Immunosuppression** | | | | | | |
| T-cell Depletion Induction | 36714 (53.31%) | 8644 (48.32%) | 8687 (49.82%) | 10102 (57.96%) | 9281 (57.61%) | <0.001 |
| IL-2RA Induction | 17001 (24.69%) | 5357 (29.95%) | 4213 (24.16%) | 3859 (22.14%) | 3572 (22.17%) | <0.001 |
| Calcineurin Inhibitor | 61085 (88.70%) | 16224 (90.70%) | 15556 (89.21%) | 15819 (90.76%) | 13486 (83.71%) | <0.001 |
| Maintenance Steroids | 47498 (68.97%) | 13055 (72.98%) | 12936 (74.19%) | 12586 (72.21%) | 8921 (55.37%) | <0.001 |
| **Donor Characteristics** | | | | | | |
| Donor Age, Years | 41.08 ± 14.95 | 40.28 ± 15.00 | 41.02 ± 15.11 | 41.23 ± 15.14 | 41.86 ± 14.48 | <0.001 |
| Donor Sex, Male | 35559 (51.64%) | 9353 (52.29%) | 9229 (52.93%) | 9063 (52.00%) | 7914 (49.12%) | <0.001 |
| AA Donor Ethnicity | 8802 (12.88%) | 1844 (10.36%) | 2593 (15.06%) | 2463 (14.22%) | 1902 (11.87%) | <0.001 |
| Missing | 505 (0.7%) | 94 (0.5%) | 218 (1.3%) | 103 (0.6%) | 90 (0.6%) | |
| Donor BMI | 27.38 ± 6.04 | 27.18 ± 5.89 | 27.42 ± 6.11 | 27.40 ± 6.16 | 27.52 ± 5.98 | <0.001 |
| Missing | 1438 (2.1%) | 225 (1.3%) | 370 (2.1%) | 285 (1.6%) | 558 (3.5%) | |
| Donor Diabetes, Y | 3509 (5.11%) | 797 (4.47%) | 947 (5.45%) | 1030 (5.93%) | 735 (4.58%) | <0.001 |
| Missing | 234 (0.3%) | 57 (0.3%) | 63 (0.4%) | 63 (0.4%) | 51 (0.3%) | |
| Donor Hypertension, Y | 14296 (20.93%) | 3480 (19.55%) | 3918 (22.59%) | 3870 (22.49%) | 3028 (18.97%) | <0.001 |
| Missing | 554 (0.8%) | 85 (0.5%) | 94 (0.5%) | 225 (1.3%) | 150 (0.9%) | |
| Donor Hep C Antibody, Positive | 1543 (3.42%) | 323 (2.61%) | 411 (3.43%) | 406 (3.42%) | 403 (4.54%) | <0.001 |
| Missing | 23768 (34.5%) | 5521 (30.9%) | 5437 (31.2%) | 5568 (31.9%) | 7242 (45.0%) | |
| Donor Don. After Cardiac Death, Y | 5764 (8.37%) | 1648 (9.21%) | 1626 (9.32%) | 1437 (8.24%) | 1053 (6.54%) | <0.001 |
| Missing | 23703 (34.4%) | 5499 (30.7%) | 5416 (31.1%) | 5551 (31.8%) | 7237 (44.9%) | |
| Donor Terminal Creatinine | 1.15 ± 0.91 | 1.07 ± 0.79 | 1.13 ± 0.87 | 1.25 ± 1.08 | 1.16 ± 0.88 | <0.001 |
| Missing | 23735 (34.5%) | 5508 (30.8%) | 5423 (31.1%) | 5559 (31.9%) | 7245 (45.0%) | |
| **Outcomes** | | | | | | |
| Graft failure (5 yr) | 7920 (11.50%) | 2107 (11.78%) | 2130 (12.22%) | 1910 (10.96%) | 1773 (11.00%) | <0.001 |
| Death (5 yr) | 9971 (14.48%) | 2737 (15.30%) | 2660 (15.25%) | 2310 (13.25%) | 2264 (14.05%) | <0.001 |
| Recipient LOS, Days | 7.60 ± 12.99 | 8.14 ± 10.48 | 7.71 ± 10.14 | 6.72 ± 13.19 | 7.85 ± 17.28 | <0.001 |
| Missing | 204 (0.3%) | 54 (0.3%) | 42 (0.2%) | 44 (0.3%) | 64 (0.4%) | |
| Follow-up (days) | 2260.46 ± 1440.88 | 2208.97 ± 1454.75 | 2261.97 ± 1430.14 | 2305.43 ± 1428.70 | 2267.32 ± 1448.41 | <0.001 |
| Treated for rejection within 1 year | 4848 (9.28%) | 1310 (9.46%) | 1272 (9.77%) | 1194 (8.82%) | 1072 (9.06%) | 0.041 |

*(Continued)*

**Table 4.** (Continued)

| | Overall (n = 68865) | Low (Q1; n = 17888) | Medium (Q2; n = 17437) | Medium-High (Q3; n = 17429) | High (Q4; n = 16111) | P-value* |
|---|---|---|---|---|---|---|
| Missing | 16626 (24.1%) | 4039 (22.6%) | 4413 (25.3%) | 3891 (22.3%) | 4283 (26.6%) | |

Notes: Continuous variables summarized as mean±standard deviation; categorical variables, as count (percentage).

"*" Testing the difference between all four groups.

"**" reported only for deceased donor kidneys.

Abbreviations: AA, African American; BMI, Body mass index; CIT, Cold ischemia time; LOS, Length of stay; TX, Transplant; IL-2RA, Interleukin-2 receptor antagonist; HLA, Human leukocyte antigen; PRA, Panel reactive antibody; Y, Yes

as well as patient death. We report that, compared to low (Q1) volume centers, medium-high (Q3) volume (median: 121 transplants/year) US transplant centers had better outcomes for DDKT. The Q1 center volume was set as the reference throughout the analysis. However, LDKT outcomes do not show significant associations with center volume.

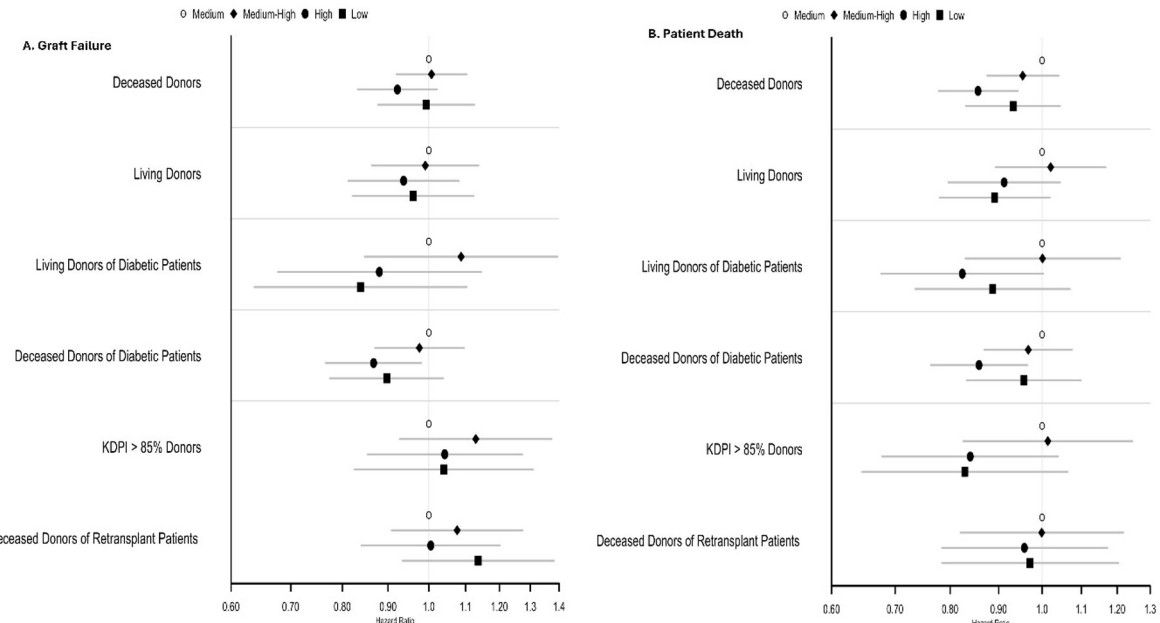

**Fig 2.** Forest Plots for Patient Outcomes Stratified by Center Volume: **(A) Adjusted Cause-specific hazard ratios (aCHR) for Five-Year Graft Failure by CRA.** Low volume center is reference standard (HR: 1.0). **Deceased Donors**: Medium-volume aCHR = 1.0005 (95% CI = 0.950–1.063; p = 0.863), Medium-high volume aCHR = 0.892 (95%CI = 0.841–0.946; p<0.001), High-volume aCHR = 0.953 (95% CI = 0.893–1.017; p = 0.149); **Living Donors**: Medium-volume aCHR = 0.968 (95%CI = 0.860–1.089; p = 0.589), Medium-high volume aCHR = 0.897 (95%CI = 0.796–1.012; p = 0.077), High-volume aCHR = 0.90 (95%CI = 0.798–1.014; p = 0.085); **Diabetic Patients**: Medium-volume aCHR = 0.96 (95%CI = 0.871–1.058; p = 0.406), Medium-high volume aCHR = 0.848 (95%CI = 0.768–0.936; p = 0.001), High-volume aCHR = 0.881 (95%CI = 0.787–0.986; p = 0.027); **KDPI>85% donors:** Medium-volume aCHR = 1.096 (95%CI = 0.914–1.314; p = 0.325), Medium-high volume aCHR = 0.999 (95%CI = 0.832–1.2; p = 0.992), High-volume aCHR = 0.997 (95%CI = 0.813–1.221; p = 0.974); **Re-transplants**: Medium-volume aCHR = 1.052 (95%CI = 0.904–1.224; p = 0.513), Medium-high volume aCHR = 0.969 (95% CI = 0.83–1.131; p = 0.686), High-volume aCHR = 1.101 (95%CI = 0.937–1.294; p = 0.241); **(B) Adjusted Cause-specific hazard ratios (aCHR) for Five-Year Patient Death by CRA**. Low volume center is reference standard (HR: 1.0). **Deceased Donors**: Medium-volume aCHR = 0.956 (95%CI = 0.899–1.017; p = 0.157), Medium-high volume aCHR = 0.828 (95%CI = 0.776–0.884; p<0.001), High-volume aCHR = 0.898 (95%CI = 0.836–0.965; p = 0.003); **Living Donors**: Medium-volume aCHR = 1.006 (95%CI = 0.881–1.150; p = 0.925), Medium-high volume aCHR = 0.895 (95%CI = 0.0.781–1.024; p = 0.107), High-volume aCHR = 0.88 (95%CI = 0.77–1.006; p = 0.061); **Diabetic Patients:** Medium-volume aCHR = 0.964 (95%CI = 0.885–1.05; p = 0.396), Medium-high volume aCHR = 0.826 (95% CI = 0.755–0.905; p<0.001), High-volume aCHR = 0.928 (95%CI = 0.84–1.025; p = 0.14); **KDPI>85% donors:** Medium-volume aCHR = 0.981 (95%CI = 0.813–1.182; p = 0.838), Medium-high volume aCHR = 0.776 (95%CI = 0.64–0.941; p = 0.01), High-volume aCHR = 0.787 (95%CI = 0.634–0.977; p = 0.03); **Re-transplants**: Medium-volume aCHR = 0.989 (95%CI = 0.812–1.204; p = 0.91), Medium-high volume aCHR = 0.94 (95%CI = 0.77–1.149; p = 0.548), High-volume aCHR = 0.963 (95%CI = 0.777–1.193; p = 0.73).

Excellence in patient health related outcomes are the fundamental goals of any health care system [24, 25]. The relationship between kidney transplant center volume and patient outcomes has been a debated issue [1]. To some degree, our results corroborate the findings of other studies in the field of solid organ transplantation including heart, lung, liver and pediatric kidney transplants where higher center volume was associated with improved outcomes [26–30]. One major limitation of these previous studies has been an examination of only short-term outcomes that were limited to 1-year post-transplant [1–3, 8]. The selection for these specific time periods is understandable as the intent of the authors was to capture the most vulnerable period after transplant, where intra-operative and post-operative care can immediately affect the clinical outcomes. A similar association is seen in almost all surgical arenas where it could be said that 'practice indeed makes perfect' [31]. The flip side of this question is the impact a high transplant center volume may have on long-term outcomes. One can conceive that high-volume programs may have high-volume physicians with a potential detrimental impact on patient care delivery due to constraints of time and staff. Necessities of transitions of care to community physicians may further result in fragmentation of care [32]. Indeed, it has been shown that increased patient volume may have a detrimental impact on dialysis outcomes, re-admission rates, cost of care, as well on preventive care delivery [5, 33–35]. Based upon the data from our study, where medium-high volume (Q3) centers achieved better outcomes, it is tempting to hypothesize that there may be a 'sweet spot' where one could derive the advantages of superior peri-operative care without compromising long-term care due to excessive patient volume. In addition, there may be other advantages with regards to reduced cost of care, as suggested by the reduced index transplant length of stay in Q3 centers across all analyzed sub-populations in our study.

Some of our findings are in contrast to the study by Sonnenberg et al who found no difference in allograft and patient survival for both DDKT and LDKT based on the center volume [8]. There are several possible explanations for this discrepancy. First and foremost is the difference in methodology. Sonnenberg et al used conventional methods of multivariate cox regression for analysis [9, 10]. These conventional models work well when the censoring event occurs independently (such as loss to follow up), however they become less accurate when the censoring event is not entirely independent i.e. death [9]. Several recent studies report the potential of erroneous conclusions when utilizing the Kaplan-Meier (KM) method to estimate probabilities of an outcome over time [9, 11, 36]. Similar to the example by El Ters et al, a case in point are the diabetic patients in our study, where the 5-year mortality is much higher compared to non-diabetics with DDKT (14.48% vs 10.83%) while graft loss is better (11.50% vs 14.36%), thus skewing the denominator for KM analyses [11]. Secondly, we used a longer period of follow up (5 years post-transplant) compared to 3 years post-transplant. Conventional methods of analysis could produce erroneous results in populations with competing risks, as with the passage of increasing lengths of time competing events are more likely to occur, and in that situation CRA may be a more appropriate approach [11]. Third, we used a much larger sample size to increase the power of our study.

The lack of association between kidney transplant center volume and both graft loss and mortality in LDKT recipients may be attributed to several factors. One plausible explanation could be that LDKT procedures are often characterized by different dynamics and considerations compared to deceased donor procedures. Living donor kidney transplants typically involve healthier organs and donors, leading to generally favorable outcomes. Additionally, LDKTs are often planned and scheduled in advance, allowing for meticulous donor and recipient matching, and optimized perioperative care. The distinct nature of LDKT, with factors such as pre-screening of donors and controlled timing, may diminish the impact of center volume on outcomes. In contrast, DDKT involves a different set of challenges and considerations.

Our study has several strengths. For the first time, we analyze the relationship between center volume and long-term clinical outcomes using a more 'biologically' appropriate CRA approach. We also include several biologically important variables including induction and maintenance immunosuppression that were excluded from previous studies and could be important to assess center outcome differences.

Our study has several limitations. The clinical severity of disease among patients could have been poorly captured in these registry data resulting in unmeasured bias and confounding. Sicker and/or 'difficult-to-transplant' patients could have clustered in high-volume centers with more resources. We also did not have access to re-admission rates, costs of care, and socio-economic factors of recipients and their access to transplantation.

In conclusion, our study provides evidence that medium-high center volume was associated with lower graft failure and all-cause mortality for adult DDKT. Further studies on these topics would be important particularly in the context of re-imbursement with the newly released Advancing American Kidney Health Initiative [37]. Irrespective though, given the benefit of transplantation over dialysis across all centers, the highest priority remains referring a patient for a kidney transplant appropriate to their level of complexity at which the patient is willing and able to receive care [38]. Further studies are required to determine the factors affecting these clinical outcomes based on center logistics, practice protocols, transitions of care, and long-term care partnership models so that best practices that be incorporated in to future long-term delivery of care for these patients.

## Supporting information

**S1 Appendix.**
(DOCX)

## Author Contributions

**Conceptualization:** Bekir Tanriover, Gaurav Gupta.

**Formal analysis:** Edem Defor, Dipankar Bandyopadhyay, Bekir Tanriover.

**Methodology:** Edem Defor, Bekir Tanriover.

**Project administration:** Ambreen Azhar, Gaurav Gupta.

**Supervision:** Ambreen Azhar, Gaurav Gupta.

**Visualization:** Ambreen Azhar, Layla Kamal.

**Writing – original draft:** Ambreen Azhar.

**Writing – review & editing:** Gaurav Gupta.

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
