## [Decision Letter · Decision Letter 0]

14 Sep 2023

PONE-D-23-11616Center Volume and Transplant Outcomes in Adult Kidney Transplant Recipients: A Competing Risk AnalysisPLOS ONE

Dear Dr. Azhar,

Thank you for submitting your manuscript to PLOS ONE. After careful consideration, we feel that it has merit but does not fully meet PLOS ONE’s publication criteria as it currently stands. Therefore, we invite you to submit a revised version of the manuscript that addresses the points raised during the review process.

Please revise.

We look forward to receiving your revised manuscript.

Kind regards,

Academic Editor

PLOS ONE

Journal Requirements:

Reviewers' comments:

Reviewer's Responses to Questions

**Comments to the Author**

1. Is the manuscript technically sound, and do the data support the conclusions?

Reviewer #1: Yes

Reviewer #2: Yes

Reviewer #3: Partly

Reviewer #4: Yes

2. Has the statistical analysis been performed appropriately and rigorously? 

Reviewer #1: Yes

Reviewer #2: No

Reviewer #3: No

Reviewer #4: No

3. Have the authors made all data underlying the findings in their manuscript fully available?

Reviewer #1: Yes

Reviewer #2: Yes

Reviewer #3: Yes

Reviewer #4: No

4. Is the manuscript presented in an intelligible fashion and written in standard English?

Reviewer #1: Yes

Reviewer #2: Yes

Reviewer #3: Yes

Reviewer #4: Yes

5. Review Comments to the Author

Reviewer #1: In this UNOS database anlysis using a competing risk model, the authors compared kdiney transplant outcomes ( 5 year graft failure and patient death) based on center volume (categorized as low, medium, medium high and high volume centers). The study found best outcome for deceased donor kidney recipients, diabetic recipients and those who received kidneys with KDPI >85% for transplants performed in medium-high volume transplant centers. The article is written well and analysis is detailed.

I have the following comments/critiques:

1. Factors such as dialysis duration before transplant, and delayed graft function after transplant can impact long term graft and patient outcomes that should be accounted for in the analysis to minimize residual bias.

2. Why was death-censored graft failure not calculated that would censor for death with functiong graft?

3. Type of health insurance, and socio economic status reflected in median household income can all imact post transplant outcomes. An attempt should be made to incorporate these variabbles in the analysis.

4. It would be nteresting to know the transplant rate (per hundred person years) between the groups. If transplant rate is low, it could be an indication that those centers were more selective in using only better quality kidneys which can translate into better outcomes.

Reviewer #2: This is a nationwide cohort study of adult kidney transplant recipients to estimate the association between kidney transplant center volume and 5-year outcomes. The authors reported that medium-high center volume was associated with improved patient and graft survival in deceased donor kidney transplant recipients. Although this is an interesting study, there are several concerns that need to be addressed.

Title

1) Based on the STROBE statement, I would suggest that the authors describe the study’s design in the title. The study’s design is more important than the statistical analysis.

Abstract

2) I would suggest that the authors ensure consistency in the methods, materials, and conclusions. The abstract should describe the results for the entire cohort, i.e., both living and deceased donor kidney transplant recipients, rather than only deceased donor kidney transplant recipients.

Introduction

3) I would suggest that the focus be on long-term follow-up as well as novelty of statistical analysis. Considering the similar previous studies, the novelty of this study lies more in the long-term follow-up than in the statistical analysis.

4) The authors stated that "We hypothesized that center volume will affect the clinical outcomes including graft failure and patient mortality." I would suggest that the authors provide a rationale for their hypothesis based on the results of the previous studies. Do the authors believe that the negative results of previous studies were due to inappropriate statistical analysis, i.e., the Cox proportional hazards model? As I mentioned above, please consider restructuring the introduction section to better clarify the novelty of the study and the rationale for the hypothesis.

Materials and Methods

5) I would suggest clarifying that the study population was "kidney-only" transplant recipients.

6) I would suggest that "Exposure" and "Outcomes" be described as separate sections.

Statistical analysis

7) I would suggest that the authors reconsider the statistical analysis, especially the subgroup analysis of kidney transplant recipients with diabetes. I understood that recipients of living and deceased donor organs were analyzed separately because of the different predicted graft longevity and because of the different covariate adjustment. This statistical approach was conducted in the previous study [1]. However, for the analysis of kidney transplant recipients with diabetes, both living and deceased donors seemed to be included in this study (Table S5). If the authors do not distinguish such recipients, then the recipients of the entire cohort should be included in the main analysis.

[1] Sonnenberg EM, Cohen JB, Hsu JY, Potluri VS, Levine MH, Abt PL, et al. Association of kidney transplant center volume with 3-year clinical outcomes. Am J Kidney Dis. 2019; 74(4):441–451.

8) I would suggest clarifying the handling of missing data. Given the results of the previous study using Organ Procurement and Transplantation Network registry data [1], I assumed that there were missing data in this study as well. Did you perform a complete case analysis? I would suggest performing the same analysis after multiple imputation of all missing covariates as a sensitivity analysis.

[1] Sonnenberg EM, Cohen JB, Hsu JY, Potluri VS, Levine MH, Abt PL, et al. Association of kidney transplant center volume with 3-year clinical outcomes. Am J Kidney Dis. 2019; 74(4):441–451.

9) Please consider restructuring the statistical analysis section to better clarify its methodology. I understood that the authors conducted competing risk analyses because the authors stated in the introduction section that "Their results show that CRA may provide more accurate estimates of long-term graft survival and death, particularly in subgroups of recipients with higher rates of competing events". In addition, the authors specified "competing risk analysis" in their title. On the other hand, the authors also stated in the statistical analysis section that "We fitted our cause-specific hazards model using the “coxph” function in the “survival” package in R." I would propose to estimate both the cause-specific hazard ratios and the subdistribution hazard ratios.

10) Given that patients who received kidney transplantation between January 2001 and December 2015 were included in this study, an adjustment by year of transplant may be necessary. In addition to immunosuppressive drugs, the quality of transplant team care and surgical techniques has evolved over time.

Results

11) When using competing risk analysis, I would suggest that the adjusted "subdistribution hazard ratio" be stated instead of the adjusted "hazard ratio" throughout the manuscript to avoid confusing the readers.

Discussion

12) I would propose to revise the text regarding the summary of results in the first paragraph of discussion section as follows: we report that, compared to low (Q1) volume centers, medium-high (Q3) volume (median: 121 transplants/year) US transplant centers had better outcomes. The Q1 center volume was set as the reference throughout the analysis.

13) I would suggest that the authors further discuss the differences in outcomes between living and deceased donor kidney transplantation. Why was kidney transplant center volume not associated with both graft loss and mortality in living donor kidney transplant recipients? Could this discrepancy be due to differences in the graft survival of living and deceased donor kidney transplant recipients?

Figure

14) In the Kaplan-Meier curve, I would suggest that the number of recipients in each group be listed at the bottom of the curve.

Table

15) I would suggest that the number of recipients in each group be listed at the top of the table. For example, Low (Q1) (n = ????).

16) As I mentioned above, I would suggest that the results of the competing analysis and the Cox proportional hazards model be described side by side. I am interested in whether the results of the two differ from each other.

Reviewer #3: 1. This research topic is interesting and worth discussing. There are significant differences in baseline variables that can affect outcome depending on the center volume. Although many variables have been statistically adjusted in the current analysis, since the center volume is a variable in which different variables are combined, it cannot be completely free from the difference between the values of the baseline variable. The statistical analysis using propensity score matching seems appropriate to fully confirm the impact of the center volume. It would be more informative to present the results of the main outcome analyzed by propensity score matching as supplementary data.

2. How many missing data was there for each variable and how was it handled during statistical analysis?

Reviewer #4: Dear Editor and authors, thank you for the opportunity to review this interesting manuscript.

This is a 5-year retrospective cohort analyzed by competing risks for graft failure and death affected by center volume. Please, see below my comments:

Major considerations

Introduction

Kindly review previously published prediction models for similar outcomes in graft loss or death kidney transplant recipients by center volume capacity or other administrative aspects.

Materials and Methods

Study population:

¿Were patients with primary renal graft thrombosis (arterial or venous) excluded?

Statistical analysis:

The methods section should clarify the modeling technique and etiology/predictor variable definition.

Please mention the performance model of this CRA analysis

It is advisable consultation of the STROBE and TRIPOD statements on reporting of observational studies and prediction models, respectively.

Does it have this model internal or external validation? please include the technique to validate or justify the lack of internal/external validation

Results

Is the increase of the survival rates in patient death with graft failure significant in the diabetic patient's group compared to patient death not from kidney graft failure? Are there differences between the other groups?

Please include the results of the internal validation

Please include the results of the performance model

Discussion

Please amplify the results of Sonnenberg et al. for a better understanding of this comparison.

Please explain the utility of this final etiology model and how it could be incorporated into some other aspects of the health system or clinical use.

Minor considerations

Discussion

The word “arenas” should be replaced by areas. Please verify. In the sentence “A similar association is seen in almost all surgical arenas where it could be said that ‘practice indeed makes perfect”

6. PLOS authors have the option to publish the peer review history of their article (what does this mean?). If published, this will include your full peer review and any attached files.

Reviewer #1: No

Reviewer #2: No

Reviewer #3: No

Reviewer #4: No

---

## [Author Response · Author response to Decision Letter 0]

7 Mar 2024

We have attached a document " Response to the reviewers to address each comment in detail". Thanks for providing us the opportunity to respond to the reviewers.

---

## [Decision Letter · Decision Letter 1]

18 Mar 2024

"Long-Term Effects of Center Volume on Transplant Outcomes in Adult Kidney Transplant Recipients."

PONE-D-23-11616R1

Dear Dr. Azhar,

We’re pleased to inform you that your manuscript has been judged scientifically suitable for publication and will be formally accepted for publication once it meets all outstanding technical requirements.

Kind regards,

Academic Editor

PLOS ONE

Additional Editor Comments (optional):

Reviewers' comments:

Reviewer's Responses to Questions

**Comments to the Author**

1. If the authors have adequately addressed your comments raised in a previous round of review and you feel that this manuscript is now acceptable for publication, you may indicate that here to bypass the “Comments to the Author” section, enter your conflict of interest statement in the “Confidential to Editor” section, and submit your "Accept" recommendation.

Reviewer #2: All comments have been addressed

Reviewer #3: All comments have been addressed

2. Is the manuscript technically sound, and do the data support the conclusions?

Reviewer #2: Yes

Reviewer #3: Yes

3. Has the statistical analysis been performed appropriately and rigorously? 

Reviewer #2: Yes

Reviewer #3: Yes

4. Have the authors made all data underlying the findings in their manuscript fully available?

Reviewer #2: Yes

Reviewer #3: (No Response)

5. Is the manuscript presented in an intelligible fashion and written in standard English?

Reviewer #2: Yes

Reviewer #3: Yes

6. Review Comments to the Author

Reviewer #2: I confirmed that the authors addressed almost all of my concerns adequately. However, I have a further additional comment. Please reflect the number of recipients in each quartile group at the top of Table 1.

Reviewer #3: The issues raised have been addressed. I have no further comments. The manuscript is ready for publication.

7. PLOS authors have the option to publish the peer review history of their article (what does this mean?). If published, this will include your full peer review and any attached files.

Reviewer #2: No

Reviewer #3: No

---

## [Editor Report · Acceptance letter]

14 May 2024

PONE-D-23-11616R1 

PLOS ONE

Dear Dr. Azhar, 

I'm pleased to inform you that your manuscript has been deemed suitable for publication in PLOS ONE. Congratulations! Your manuscript is now being handed over to our production team.

Kind regards, 

on behalf of

Dr. Robert Jeenchen Chen 

Academic Editor

PLOS ONE